# Pathway from TDP-43-Related Pathology to Neuronal Dysfunction in Amyotrophic Lateral Sclerosis and Frontotemporal Lobar Degeneration

**DOI:** 10.3390/ijms22083843

**Published:** 2021-04-08

**Authors:** Yuichi Riku, Danielle Seilhean, Charles Duyckaerts, Susana Boluda, Yohei Iguchi, Shinsuke Ishigaki, Yasushi Iwasaki, Mari Yoshida, Gen Sobue, Masahisa Katsuno

**Affiliations:** 1Institute for Medical Science of Aging, Aichi Medical University, Aichi 480-1195, Japan; iwasaki@sc4.so-net.ne.jp (Y.I.); myoshida@aichi-med-u.ac.jp (M.Y.); 2Department of Neurology, Nagoya University, Nagoya 744-8550, Japan; iguyo@med.nagoya-u.ac.jp (Y.I.); shinsuke.ishigaki@gmail.com (S.I.); ka2no@med.nagoya-u.ac.jp (M.K.); 3Department of Neuropathology Raymond Escourolle, Groupe Hospitalier Pitié-Salpêtrière Charles Foix, AP-HP-Sorbonne Université, F-75013 Paris, France; danielle.seilhean@upmc.fr (D.S.); charles.duyckaerts@aphp.fr (C.D.); susana.boludacasas@aphp.fr (S.B.); 4Faculty of Medicine, Sorbonne University, F-75013 Paris, France; 5Paris Brain Institute (ICM), Inserm U 1127, CNRS UMR7225, Sorbonne University, F-75013 Paris, France; 6Graduate School of Medicine, Aichi Medical University, Aichi 480-1195, Japan; sobueg@aichi-med-u.ac.jp

**Keywords:** ALS, autophagy, FTLD, synapse, TDP-43

## Abstract

Transactivation response DNA binding protein 43 kDa (TDP-43) is known to be a pathologic protein in amyotrophic lateral sclerosis (ALS) and frontotemporal lobar degeneration (FTLD). TDP-43 is normally a nuclear protein, but affected neurons of ALS or FTLD patients exhibit mislocalization of nuclear TDP-43 and cytoplasmic inclusions. Basic studies have suggested gain-of-neurotoxicity of aggregated TDP-43 or loss-of-function of intrinsic, nuclear TDP-43. It has also been hypothesized that the aggregated TDP-43 functions as a propagation seed of TDP-43 pathology. However, a mechanistic discrepancy between the TDP-43 pathology and neuronal dysfunctions remains. This article aims to review the observations of TDP-43 pathology in autopsied ALS and FTLD patients and address pathways of neuronal dysfunction related to the neuropathological findings, focusing on impaired clearance of TDP-43 and synaptic alterations in TDP-43-related ALS and FTLD. The former may be relevant to intraneuronal aggregation of TDP-43 and exocytosis of propagation seeds, whereas the latter may be related to neuronal dysfunction induced by TDP-43 pathology. Successful strategies of disease-modifying therapy might arise from further investigation of these subcellular alterations.

## 1. Introduction

Transactivation response DNA binding protein 43 kDa (TDP-43) is known to be a pathologic protein in amyotrophic lateral sclerosis (ALS) and frontotemporal lobar degeneration (FTLD) [1,2,3]. TDP-43 is an RNA-binding protein, which is coded on chromosome 1p [3]. Systemic organs, including the central nervous system, pancreas, and spleen, abundantly express TDP-43, but its physiological function is largely unknown [3]. TDP-43 is localized in the nucleus in normal settings, but affected neurons of ALS or FTLD patients exhibit mislocalization of nuclear TDP-43 and cytoplasmic inclusions. Pathological and molecular studies have suggested either gain-of-neurotoxicity of aggregated TDP-43 or loss-of-function of intrinsic, nuclear TDP-43. However, a mechanistic discrepancy between the TDP-43 pathology and neuronal dysfunctions remains. Recently, basic researches have reported that abnormalities in TDP-43 are associated with dynamic and complex alterations of neuronal substructures and metabolism. Concordant neuropathologic evidence has also been accumulated from postmortem patient studies of TDP-43-related ALS or FTLD (ALS-TDP and FTLD-TDP, respectively). This review aims to discuss pathways from TDP-43 pathology to mechanisms leading to neuronal dysfunction.

## 2. Clinical Findings of ALS and FTLD

ALS encompasses a sporadic or familial motor neuron disease, which is clinically characterized with upper and lower motor neuron signs and symptoms [4]. The muscle weakness is relentlessly progressive and lethal. The median survival duration is about three years from disease onset, and the average age of onset is 58–60 years [5]. The prevalence is approximately six cases per 100,000 [5]. Several clustered regions are known to have a high prevalence of ALS, including Guam island of USA, Kii peninsula of Japan, and West New Guinea. Patients in the clustered foci often show a phenotype of parkinsonism-dementia complex (ALS/PDC) that is atypical of classical ALS. The prevalence has recently decreased in these regions for unknown reasons, although it is still high in New Guinea [6].

FTLD is the pathological term corresponding to the clinical term of frontotemporal dementia (FTD) [7]. FTD is the second most common form of dementia after Alzheimer-type dementia. A study from the UK reported that the prevalence of FTLD was 10.8 per 100,000 population and highest between 60 and 69 years, although the data set also included progressive supranuclear palsy (PSP) and corticobasal syndrome [8]. FTD is subclassified into behavioral variant FTD (bvFTD) [9], progressive non-fluent aphasia (PNFA) [10], and semantic dementia (SD) [10]. bvFTD is characterized by disinhibition and executive disorders and more common than PNFA and SD. PNFA and SD manifest as impairment of output and input of language, respectively.

## 3. TDP-43 Pathology in ALS and FTLD

There is a loss of upper and lower motor neurons in ALS patients, which results in regional atrophy. Atrophy of the anterior roots is the most informative finding upon macroscopic observations, whereas that of the primary motor cortex is usually mild (Figure 1). TDP-43 pathology has been observed in 95% of sporadic ALS cases [11,12], followed far behind by fused-in-sarcoma (FUS) [13,14]. TDP-43 is mislocalized from the nucleus and aggregates in the cytoplasm of motor neurons in ALS patients (Figure 2). Aggregated TDP-43 is phosphorylated, ubiquitylated, and truncated at the C-terminal side [15,16]. The TDP-43 pathology is more prominent in the lower motor neurons in the spinal cord and cranial nerve nuclei than in the upper motor neurons (Betz cells) of the primary motor cortex. Although eye movement, sensation, and urorectal functions are spared in ALS patients, TDP-43 pathology has occasionally been found in the oculomotor nerve nucleus, Clarke column, and Onuf nucleus [17,18]. Another finding of importance in ALS is Bunina bodies, which are eosinophilic inclusion bodies found in the neuronal cell body. Bunina bodies are an accumulation of tubular and vesicular structures from unknown origin and do not represent a cytoplasmic aggregation of TDP-43, although a subset of those demonstrates immunoreactivity to TDP-43 [19,20].

In FTLD, TDP-43 and tau each accounts for nearly 50% [21]. FTLD-TDP is currently subclassified into pathological types A, B, and C [22,23,24]. Type A is characterized by TDP-43-immunoreactive, short dystrophic neurites, and cytoplasmic inclusions in the upper cortical layers. In type B, crescent or ring-shaped cytoplasmic aggregation of TDP-43 are observed across all cortical layers. Type C is characterized by TDP-43 immunopositive thick and long dystrophic neurites in the upper cortical layers, and cytoplasmic inclusions are rare. The frontotemporal neocortices are vulnerable, and the hippocampus, the amygdala, the neostriatum, and the substantia nigra are also preferentially involved with TDP-43 pathology [25,26] (Figure 2). Cortical TDP-43 pathology often spreads toward more posterior areas than prefrontal areas, in contrast to the topography of Pick disease lesions. The distribution of FTLD lesions largely corresponds to clinical phenotypes of FTD: involvement of frontal and temporal cortices is seen in bvFTD; that of the anterior temporal cortices is seen in SD, and that of the frontal cortices and para-Sylvian areas is seen in PNFA [27] (Figure 1).

ALS-TDP and FTLD-TDP may overlap clinically or pathologically [28]. ALS-TDP and FTLD-TDP often coexist in the same individuals; this phenotype is currently termed frontotemporal dementia with motor neuron disease (FTD-MND). Regions vulnerable to the TDP-43 pathology are often the same in ALS-TDP and FTLDTDP. For example, TDP-43 aggregations in the hippocampus were found in 40% of the non-demented ALS-TDP patients, whereas TDP-43 aggregations in the spinal cord motor neurons were found in 90% of the FTLD-TDP patients even in the absence of motor neuron signs or symptoms [29,30]. It has been suggested that ALS-TDP and FTLD-TDP could be a part of a continuous spectrum of diseases, ‘TDP-43 proteinopathy’ [28]. However, molecular assays of aggregated TDP-43 have revealed different molecular properties between ALS-TDP and FTLD-TDP and among pathological subtypes of FTLD-TDP. The molecular weights of C-terminal fragments of aggregated TDP-43 differ among FTLD-TDP type A, B, and C, and ALS [15,31]. In addition, a recent study revealed that molecular size, density, structure, and neurotoxicity differ among the subtypes of TDP-43 proteinopathy [32]. This evidence indicates the possibility of a distinctive, at least partially, molecular pathway of TDP-43 aggregation in each pathological phenotype of TDP-43 proteinopathy.

## 4. Gain-of-Neurotoxicity and Loss-of-Function

These ambivalent concepts may have arisen from the double face of TDP-43 pathology: cytoplasmic aggregation and mislocalization from the nucleus. It is broadly believed that the cytoplasmic inclusions of TDP-43 are neurotoxic. Neuronal death or axonal dysfunction has been observed in models with overexpression of TDP-43 [33] and cells transfected with pathological TDP-43 mutants [34,35] or cytoplasmically mislocalized TDP-43 with mutated nuclear localization signals (NLSs) [36]. How aggregated TDP-43 triggers neuronal death or dysfunction remains controversial. By contrast, evidence of loss-of-function of TDP-43 mechanisms has also been accumulated. Transgenic mice expressing human TDP-43 with a mutated NLS displayed neuronal loss and tract degeneration. Endogenous nuclear TDP-43 is downregulated, and cytoplasmic inclusions were sparse. These facts suggest that the loss of nuclear TDP-43 is more strongly correlated with neuronal dysfunction than is the cytoplasmic inclusions [37]. Other TDP-43 suppression or knock-out models also showed neuronal dysfunction, including an alteration of TDP-43-related transcriptome resulting in synaptic abnormality [38], deficits in DNA repair [39], a loss of splicing repressor function [40], and dsRNA-foci [41].

## 5. Anatomical Spreading of TDP-43 Pathology

Neuropathologic analyses of autopsied patients with ALS-TDP and FTLD-TDP have indicated that TDP-43 pathology spreads along certain neurally connected anatomical systems rather than depending on spatial proximity. Direct or indirect corticofugal spreading of TDP-43 pathology from the primary motor cortex to the lower motor neurons has been suggested in ALS-TDP patients [42,43]. In FTLD-TDP, cortico-cortical spreading from prefrontal areas to caudal cortices has been proposed [44]. Subcellular observations revealed granular aggregates of phosphorylated TDP-43 (p-TDP-43) within the axons and axonal terminals of ALS-TDP and FTLD-TDP patients, which indicates an intra-axonal transfer of the aggregates [43,45,46]. This finding is often observed in patients with short clinical duration (Figure 3) [47]; hence, p-TDP-43 aggregations might be transferred through the axons even in the early stages of the disease.

Premortem neurophysiological studies have supported a corticofugal manner in the spreading of ALS lesions. Studies of transcranial magnetic stimulation revealed that reduction in short-interval intracortical inhibition precedes lower motor neuron dysfunction among ALS patients, indicating early impairment of intracranial circuit within the primary motor cortex [48]. Structural [49,50,51] and functional [52] imaging technics have also suggested the corticofugal manner of ALS lesions; for instance, a study of resting-state functional MRI showed that patterns of increased connectivity, relevant to network expansion and physical disability, were consistent with spreading patterns of TDP-43 pathology in ALS [52].

Other patterns in the spreading of TDP-43 pathology have also been suggested. A neuropathological study suggested that the dentate gyrus of the hippocampus is a starting point for TDP-43 pathology in ALS-TDP and FTD-MND; the pathology may subsequently spread to the anterior olfactory nucleus, the periamygdaloid complex, the piriform cortex and eventually reach the orbital cortex and olfactory bulb [53,54]. Moreover, TDP-43 aggregates and dipeptide repeat proteins systematically involve the circadian sleep/wake-associated regions, including the pineal body and hypothalamic neurons related to the suprachiasmatic nucleus, in ALS patients with *C9orf72* repeat expansion [55].

Molecular biological studies have revealed propagative properties of TDP-43 aggregates. Full-length TDP-43 contains NLSs, whereas the C-terminal fragments do not [16]. Therefore, the fragments are considered to be more often present in the cytoplasm than full-length sequences and are thus good candidates as ‘seeds’ for aggregations [16,36,37]. Induction of patient-derived TDP-43 seeds resulted in the spreading of TDP-43 aggregates in a cell-to-cell and in a corticofugal manner for SH-SY5Y cells and mice with a mutated human TDP-43 NLS (CamKIIa-hTDP-43NLSm), respectively [16,56].

## 6. Are ALS-TDP and FTLD-TDP Synaptopathies?

Ultrastructural observations of autopsied ALS patients revealed degradation of axon terminals in the motor neuron system, associated with alterations of pre-synaptic vesicles, mitochondria, and neurofilament bundles [57,58]. We have found loss of axon terminals in cortico-striatal projections, striatopallidal projections, and striatonigral projections in autopsied ALS-TDP and FTLD-TDP patients using immunolabeling of glutamatergic or GABAergic pre-synaptic vesicles [45,46] (Figure 3). The depletion of axon terminals was more severe in FTLD-TDP patients than in ALS-TDP patients without dementia. Aggregation of p-TDP-43 was also found within the remaining terminals. A recent study quantified axonal terminal densities with high precision using array tomography. It revealed significant depletion of axonal endings in the prefrontal cortex and its correlation with cognitive decline in ALS patients [59].

Studies using models of increased expression of TDP-43 have reported a loss of axon terminal or dendritic spines. It has also been suggested that TDP-43 is physiologically transported within the axons and contributes to axonal outgrowth [60]. Mice overexpressing human-TDP-43 showed reductions in the expression of mRNAs that encode proteins involved in pre-synaptic activity via an RNA-binding property of TDP-43 [61]. It is unknown whether the aggregation of p-TDP-43 within axonal terminals demonstrates in situ synaptotoxicity, which has been reported for a tauopathy model; induction of pathogenic mutant tau bound to pre-synaptic vesicles and disrupted synaptic functions [62]. Depletion of TDP-43 in neurons or microglia has also been reported to be correlated with synaptic loss, impairment of synaptic plasticity, alterations in RNA transcripts that are closely relevant to synaptic plasticity, or disruption of axonal growth [38,63,64,65]. We also found that FUS, which is the second most common pathological protein in ALS, plays a role in synaptic functions. Downregulation of FUS expression by shRNA resulted in depletion of dendritic spines and AMPA receptors and a loss of spine plasticity in primary cortical neurons and mice, respectively [66]. Synaptopathy might explain the neuronal dysfunction in a broad spectrum of disorders related to ALS and FTLD.

## 7. Impaired Clearance of TDP-43

Aggregated TDP-43 is ubiquitylated and tagged with p62, which indicates a contribution of ubiquitin-proteasome and endosome-autophagy systems to process the aggregates. The role of p62 would be to guide the ubiquitylated proteins towards the autophagy system [67]. A neuropathological study reported that a subset of TDP-43 aggregates is immunopositive for LC3 [68], which is an autophagosomal marker [60]. Activation of the autophagosome-lysosome system has been reported to decrease the neurotoxicity of aggregated TDP-43 in neuron models [69,70]. In a yeast model, increased TDP-43 level disrupted the fusion and function of vesicles linked to the autophagy-lysosome system [71]. In addition, we have shown the importance of exosomes, which are another endosome-mediated clearance mechanism in the metabolism of TDP-43 (Figure 4). Neuro2a cells that were transfected with human-strain TDP-43 secreted the exogeneous TDP-43 with the exosomes [72]. Moreover, high levels TDP-43 and insoluble C-terminal fragments were found in the exosome fraction, which was extracted from FTD-MND patients’ brains. Inhibition of exosomes resulted in intracytoplasmic mislocalization and aggregation of TDP-43. Intake of exosome carrying C-terminal fragments into the cells also facilitated the TDP-43 pathology. These facts indicate that exosomal secretion of TDP-43 is a critical step in TDP-43 metabolism and that secreted TDP-43 functions as a seed for neuron-to-neuron propagation of TDP-43 [16,73]. When considering together with a concept of synaptopathy, it can be hypothesized that transmission of TDP-43 seeds between pre- and post-synapse, depending on the exocytosis, could be a mechanism of centrifugal, anatomical propagation of TDP-43 pathology. Exosomal fraction is also identified in plasma and cerebrospinal fluid (CSF). However, whether exosomal TDP-43 levels are elevated there in patients with ALS is controversial [74].

ALS-TDP- or FTLD-TDP-related genes, including *SQSTM1* (*p62*), *UBQLN2*, *VCP*, *GRN*, and *OPTN*, are important players in the autophagy system [75]. Indeed, homozygous mutations of the *GRN* gene cause neuronal ceroid lipofuscinosis, which is characterized by lysosomal storage of lipopigment [76]. Haploinsufficiency of *C9orf72* correlated to disruption of autophagy, particularly of endosomal trafficking [77]. Upon our postmortem observations, the hippocampi and frontal cortices of *C9orf72*-mutated patients frequently and broadly displayed granulovacuolar degeneration (GVD) containing an accumulation of CHMP2B (charged multivesicular body protein 2B) [78]. CHAMP2B is a marker of the multivesicular body, which is the turning point of the lysosome or exosome pathway. CHMP2B-immunopositive vesicles of GVD often coexisted with dipeptide repeat proteins, derived from non-ATG-dependent translation of repeat sequences [78,79]. A mutation in the *CHMP2B* gene is also known to cause familial FTD, although the postmortem findings remain unknown [80]. A recent study reported that GVDs carrying necrosome-markers were primarily found in the hippocampus of ALS-TDP and FTLD-TDP with and without *C9orf72* mutation [81].

It is well known that chaperone proteins are critical to regulating the folding or solubility of pathologic proteins in motor neuron diseases and other neurodegenerative disorders [82,83]. A comprehensive study has revealed that the heat-shock response reduced insoluble, phosphorylated TDP-43 and was suppressed in spinal cord tissues of sporadic ALS-TDP patients [84]. It has also been reported that chaperon-mediated autophagy (CMA), which specifically directs the degradation of soluble proteins, regulates TDP-43 metabolism under physiological and pathologic conditions and that aggregated TDP-43 affects the integrity of CMA-associated lysosomes [85]. Degradation of liquid-liquid phase separation along with aging, genetic, or environmental factors, followed by reduction in HSP-70, has been reported to perturb the solubility of TDP-43 [86]. A postmortem study hypothesized that abnormal but soluble TDP-43 in the Betz cells, where TDP-43 pathology is considered to be initiated, could be an early mechanism of ALS; the hypothesis arose from the fact that aggregation of p-TDP-43 was relatively sparse in the Betz cells compared with mislocalization of nuclear TDP-43 [87]. Full-length TDP-43, the 45 kDa form, and ubiquitylated TDP-43 have been found in the soluble, inclusion-free fraction of brain tissues from ALS-TDP patients, which indicates that mislocalization of TDP-43, preceding the aggregation, play a role in the early pathogenesis of ALS [88].

In addition to the intraneuronal machinery, neuroinflammation toward TDP-43 aggregation has been emphasized. A study revealed high extracellular expression of a neuroinflammatory marker, cyclophilin-A, in CSF from *SOD-1*^G93A^ models and sporadic ALS patients, and drug-derived inhibition of cyclophilin-A reduced nuclear factor kappa B (NF-κB) activation, endoplasmic reticulum stress, and insoluble phosphorylated TDP-43 [89]. From the viewpoint of neuroinflammation, the pathophysiological importance of glial reactions increases. Recent studies revealed neuroprotective properties of astrocytes and microglia, of which activation and proliferation reduce pathological TDP-43 levels [90,91]. Impacts of altered glial expression of TDP-43 have also been investigated. A study of primary-cultured astrocytes suggested that knock-down of TDP-43 facilitated neuroinflammatory along with elevated NF-κB and dsRNA-foci [92], whereas a contradictory study described no influences of it toward astrocytic activation and neuronal survival [93].

## 8. Linkage between TDP-43 and tau Pathology

Postmortem studies have demonstrated that TDP-43 pathology is observed in a subset of non-ALS/FTLD disorders, including Alzheimer disease (AD) [94,95], PSP [96], corticobasal degeneration (CBD) [97], Lewy body disease (LBD) [98,99,100,101], hippocampal sclerosis [102], *LRRK-2* mutated Parkinson disease [103], and post-traumatic chronic encephalopathy [104]. Subpopulations (up to 20%) of neurologically healthy, aged people also show TDP-43 aggregation in the limbic systems, which is referred to as limbic-predominant age-related TDP-43 encephalopathy (LATE) [105].

The prevalence of TDP-43 pathology is particularly high in AD patients and accounts for up to 75% [94,95]. Impacts on clinical manifestations or on brain atrophy of the comorbid TDP-43 pathology in AD, PSP, and CBD patients have been reported [94,96,97], although whether the comorbid TDP-43 pathology is ‘bystander’ or ‘pathogenic’ is still controversial [106]. The molecular basis of comorbid TDP-43 pathology in miscellaneous disorders remains unclear. At least, the spatial distribution, the spreading pattern, and the morphology of TDP-43 aggregations partially but definitely differ from those observed in patients with ALS-TDP or FTLD-TDP. This is in great contrast with the reported similarities between AD-related tau pathology and primary age-related tauopathy [107].

Additionally, abnormalities in tau protein have also been discussed in ALS-TDP patients. Aggregation or hyperphosphorylation of tau has been observed in the motor neurons of ALS-TDP patients [108,109]. It has been reported that FTLD-TDP patients carrying *C9orf72*-repeat expansions tended to have a higher burden of tau aggregations in the temporal cortex and hippocampus than *GRN*-mutated patients [110]. However, contradictory results have also been reported; a cohort-based study of autopsied FTLD/ALS-TDP and FTLD-tau indicated no predisposition towards TDP-43 pathology in FTLD-tau patients or to tau pathology in FTLD-TDP patients [111]. ALS/PDC patients of Kii peninsula, Japan, often exhibit multiple proteinopathies, including TDP-43, tau, and α-synuclein, prominently in the limbic system (Figure 2); interestingly, the aggregation is usually mild in the motor neurons [112]. Although the genetic background and pathogenesis of this phenotype are still unknown, recent neuropathological studies have revealed that mutant ubiquitin (UBB^+1^) is highly expressed in the hippocampus or cerebral cortices, and dyshomeostasis of the ubiquitin-proteasome system is suggested [113].

The coexistence of TDP-43 and tau pathologies suggests overlapped mechanisms between these two major groups among neurodegenerative disorders. Our observations revealed that the hippocampal and cortical neurons of autopsied patients with ALS/FTLD-TDP, ALS-FUS, PSP, and CBD showed aberrant interaction of two intranuclear proteins, namely, FUS and splicing factor proline/glutamine-rich proteins (SFPQ) [114,115]. FUS and SFPQ are physiologically co-localized within the neuronal nuclear matrix, whereas they are spatially and biochemically dissociated in disease. Importantly, the dissociation of FUS and SFPQ was not observed in AD or Pick disease patients and neurologically healthy controls. An investigation using human mutant-TDP-43-knock-in mice showed alterations in splicing of *microtubule-associated protein tau (MAPT)* gene [116]. Genome analyses revealed shared genetic risks across PSP, CBD, FTD, and FTD-MND [117,118]. Intermediate repeat expansion in *C9orf72* has recently been reported to be a risk factor for CBD [119]. Upon postmortem observations of AD or CBD brains, intracellular TDP-43 aggregates and tau aggregates sometimes seem to be co-localized and sometimes not [101,120]. It remains to be clarified whether the coexistence of TDP-43 and tau pathology signifies a common mechanism upstream of pathogenesis or a process secondary to aggregation of either protein.

## 9. Conclusions

Recent studies have revealed correlations between TDP-43 abnormalities and impairment of some cellular substructures. We focused on the involvement of the endosome-autophagosome system and synaptic integrity as key actors in the pathogeneses of TDP-43 proteinopathy from the viewpoint of translational approaches across neuropathological and basic investigations. The concept of multiple proteinopathies, including interactions between TDP-43 and tau, suggests a pathophysiological link at a very early stage before protein aggregation. Although the whole pathway leading to neuronal dysfunction remains unclear, successful strategies of disease-modifying therapy may arise from those results.

## Figures and Tables

**Figure 1 ijms-22-03843-f001:**
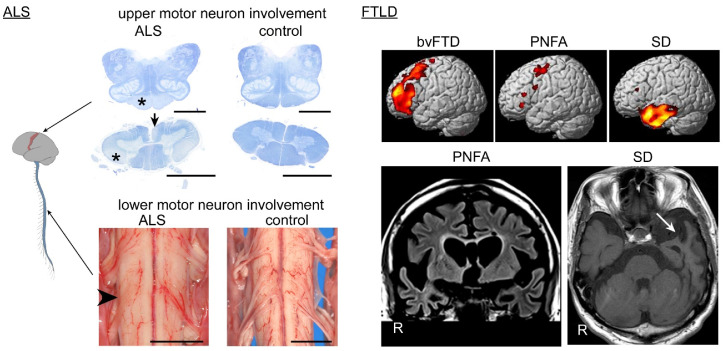
Systemic atrophy of central nervous system in amyotrophic lateral sclerosis (ALS) and frontotemporal lobar degeneration (FTLD) patients. The spinal cord and medulla oblongata are stained with Klüver–Barrera method. Involvement of the upper motor neurons results in tract degeneration of the pyramidal tract in the medullary pyramid and lateral column (*) and anterior cerebrospinal fasciculus (arrow) of the spinal cord; the change is usually prominent in the caudal segments of the spinal cord. Involvement of the lower motor neurons results in atrophy of the anterior roots in the spinal cord (arrowhead); the anterior roots are thin and hardly visible, compared with the dorsal roots. Scale bars = 5 mm. Cerebral MRI illustrates a vulnerable region corresponding to each clinical subtype: the prefrontal area for behavioral variant frontotemporal dementia (bvFTD), the para-Sylvian operculum and primary motor cortex for progressive non-fluent aphasia (PNFA), and the anterior portion of the unilateral (dominant hemisphere) temporal lobe for SD (arrow).

**Figure 2 ijms-22-03843-f002:**
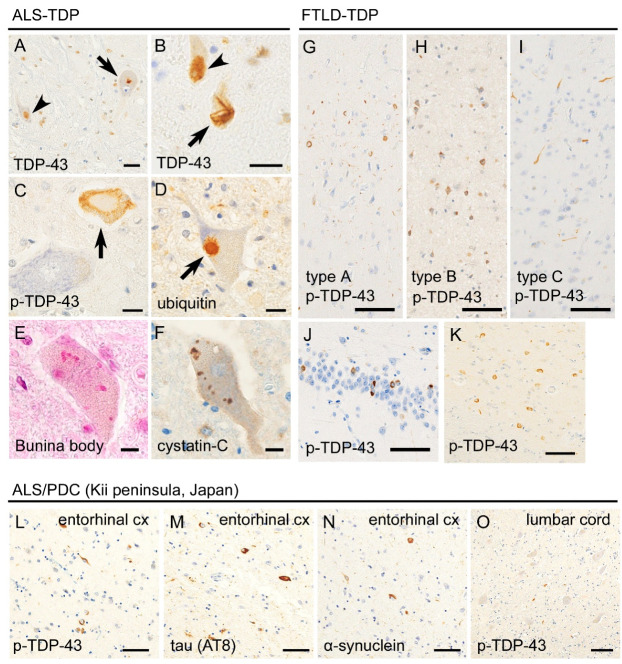
Histopathological findings of transactivation response DNA binding protein 43kDa (TDP-43)-related ALS (ALS-TDP) and FTLD (FTLD-TDP). Panels A–D were taken from an ALS-TDP patient. Anti-TDP-43 immunohistochemistry revealed that TDP-43 was mislocalized from the nucleus to the cytoplasm and forms dot-like (**A**, arrow) or skein-like inclusions (**B**, arrow) in the spinal motor neuron. Unaffected neurons showed nuclear localization of TDP-43 (**A** and **B**, arrowheads). Anti-phosphorylated TDP-43 (p-TDP-43) immunohistochemistry revealed pathologic inclusions (**C**, arrow) but not the normal nuclear expression of TDP-43. TDP-43 inclusions were immunopositive for ubiquitin (**D**, arrow). Bunina bodies are also observed in the motor neurons of ALS-TDP patients (**E**) and immunolabeled with cystation-C (**F**). Panels **G**–**K** were taken from FTLD-TDP patients. Sporadic FTLD-TDP is classified into types A, B, and C; type A is characterized by short dystrophic neurites and round- or crescent-shaped neuronal inclusions in the superficial layers of the cerebral cortex; type B is characterized by ring-shaped neuronal inclusions across all cortical layers; and type C is characterized by long and thick immunopositivity of neurites in the superficial cortical layers (**G**–**I**). Hippocampal granule cells (**J**) and neostriatum (**K**) are also preferentially involved. Panels (**L**–**O**) were taken from an ALS/ parkinsonism-dementia complex (PDC) (Kii peninsula) patient. The entorhinal cortex (**L**–**N**) showed multiple proteinopathies, including p-TDP-43 (L), hyperphosphorylated tau (**M**), and α-synuclein (**O**). Relatively mild aggregation of p-TDP-43 was observed in the spinal cord, compared with classical ALS. Scale bars: (**A**,**B**) 20 μm, (**C**–**F**) 10 μm, and (**G**–**O**) 50 μm.

**Figure 3 ijms-22-03843-f003:**
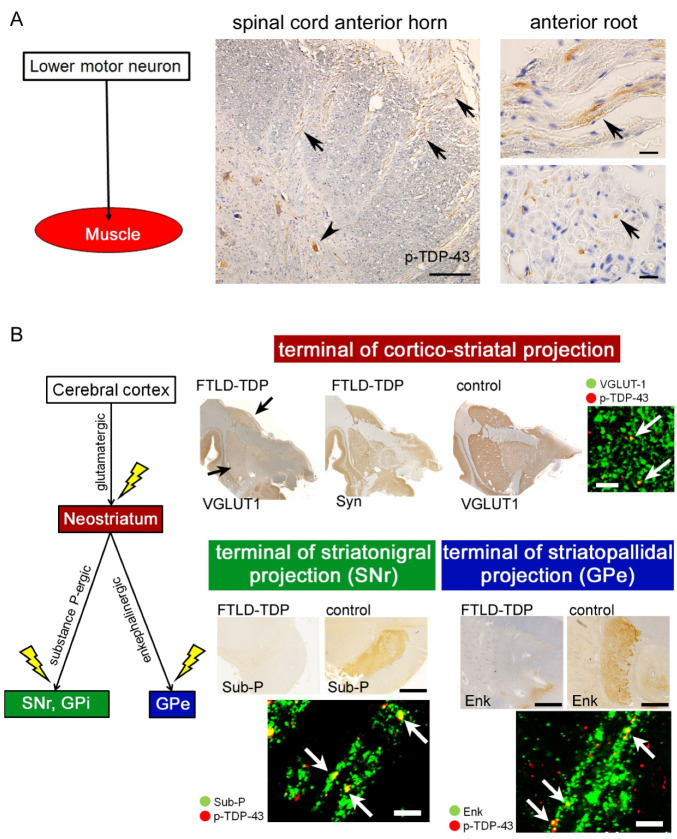
TDP-43 pathology in multi-system axons and axon terminals. The upper section (**A**) demonstrates the spinal cord of an ALS-TDP patient who died six months after the disease onset. Phosphorylated TDP-43 (p-TDP-43) aggregated not only in the anterior horn neurons (arrowhead) but also in the anterior roots (arrows). Scale bars: 100 μm for the panel of the anterior horn and 10 μm for the panels of the anterior roots. The lower section (**B**) displays pathologic changes of the cortico-striatal circuit in FTLD-TDP patients. Axon terminals of the corticofugal neurons were visualized with anti-VGLUT-1 immunohistochemistry (IHC) in the neostriatum. Those of the striatofugal neurons were labeled with anti-enkephalin (Enk) IHC in the external segments of the globus pallidus (GPe) or with anti-substance-P (Sub-P) IHC in the internal segment of GP (GPi) and pars reticulata of the substantia nigra (SNr). Patients with FTLD-TDP displayed loss of those axon terminals and p-TDP-43 aggregation within the pre-synaptic buttons. Comparing the loss of VGLUT-1-immunopositive terminals in the neostriatum (arrows) and sparing of synaptophysin (Syn) immunostaining indicates specific loss of cortico-striatal projections but spares of other projections. Scale bars: 10 μm.

**Figure 4 ijms-22-03843-f004:**
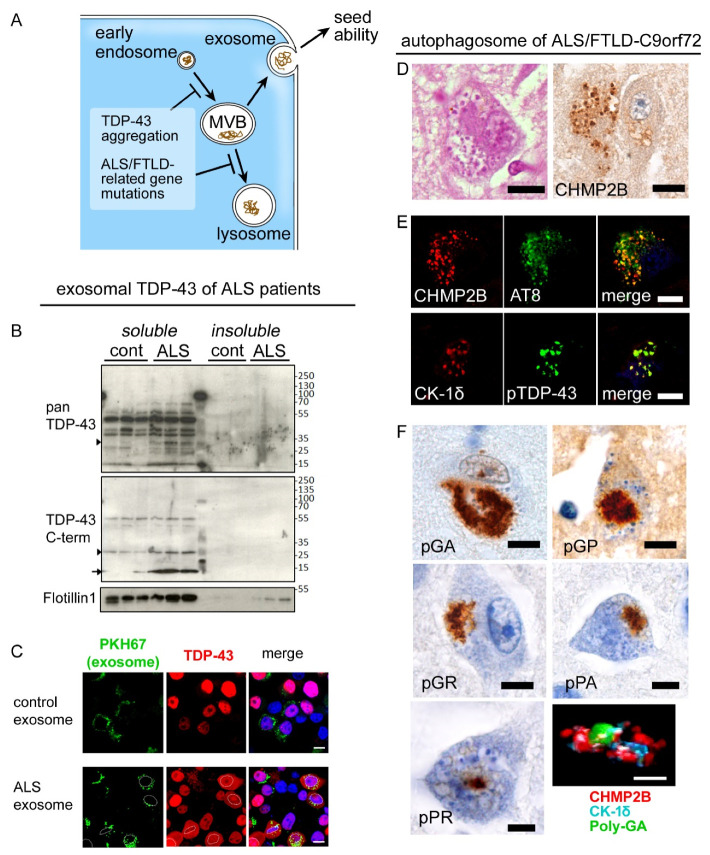
Endosome-autophagosome system and TDP-43 pathology. (**A**) Aggregated TDP-43 and dysfunction/haploinsufficiency of ALS/FTLD-TDP-related genes have been reported to impair maturation, transport, or fusion of endosomal and autophagosomal vesicles. (**B**) The exosomal fraction of brain lysates from ALS-TDP patients contains abundant TDP-43, particularly C-terminal fragments. (**C**) Neuro2a cells that were treated with ALS-patients-derived exosome and transfected with human-derived TDP-43 exhibited cytoplasmic aggregation of TDP-43. (**D**–**F**) These panels show neuropathologic changes of ALS/FTLD-TDP patients carrying C9-orf72 hexanucleotide expansions. Hippocampal pyramidal neurons often displayed granulovacuolar degeneration (GVD) that was associated with immunoreactivity for CHMP2B, a marker of multivesicular bodies (MVBs) (**D**). GVD granules, which were immunolabeled with CHMP2B and CK1δ, contained p-TDP-43 and hyperphosphorylated tau (**E**). Mutation-derived sense (poly GA, poly GP, and poly GR) and antisense dipeptides (poly PR and poly PA) were frequently covered with CHMP2B-immunopositive GVD granules. Scale bars: 10 μm.

## Data Availability

No new data were created or analyzed in this study. Data sharing is not applicable to this article.

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
