# Peer review of "Pathway from TDP-43-Related Pathology to Neuronal Dysfunction in Amyotrophic Lateral Sclerosis and Frontotemporal Lobar Degeneration"

_ijms, 2021, doi:10.3390/ijms22083843_

Round 1

Reviewer 1 Report

In this review Riku et al., addressing the pathological role of Transactivation response DNA binding protein 43 kDa (TDP-43) in ALS and FTD, discuss different pathways leading from TDP-43 pathology to neuronal dysfunction. Starting from the histopathological findings derived from post-mortem studies, they summarize the main molecular mechanisms related to TDP-43 pathology (gain of neurotoxicity and loss of function, cell-to-cell spreading, alteration of synapses function, impaired clearance of TDP-43 aggregates). In conclusion, a possible link between TDP-43 and tau pathology is discussed.

The review is carefully written and effectively summarizes the crucial findings about the main topic of the manuscript.  I have only minor suggestions.

p.5, lines 178-180 “Induction of patient-derived TDP-43 seeds resulted in spreading of TDP-43 aggregates in a cell-to-cell and in a corticofugal manner for SH-SY5Y cells and mice with a mutated human TDP-43 NLS (CamKIIa-hTDP-43NLSm), respectively”. This corticofugal spreading model (Braak et al., Nat Rev Neurol. 2013, 9, 708-714; Brettschneider J. et al., Ann Neurol. 2013, 74, 20-38) has been also confirmed in vivo by neuroimaging studies using structural (Schmidt et al., Neuroimage 2017;124:762-769; Müller et al., J Neurol Neurosurg Psychiatry 2016;87:570-9; Trojsi et al., PLoS One 2015;10:e0119045) and functional (Schulthess et al., Sci Rep. 2016;6:38391) neuroimaging techniques. A brief reference to this point could add emerging clinical information to the topic.

p. 8, lines 249-252 With regard to the impaired clearance of TDP-43, the Authors reported that "haploinsufficiency of C9orf72 correlated to disruption of autophagy, particularly of endosomal trafficking, and the hippocampi and frontal cortices of C9orf72-mutated patients frequently and broadly displayed granulovacuolar degeneration”. However, the hypothalamic-pituitary axis may show TDP-43 aggregates and dipeptide repeat protein (DPR) inclusions in C9orf72-related ALS and/or FTLD-TDP cases (Dedeene L, et al. Circadian sleep/wake-associated cells show dipeptide repeat protein aggregates in C9orf72-related ALS and FTLD cases. Acta Neuropathol Commun. 2019;7(1):189. doi: 10.1186/s40478-019-0845-9). This evidence is related to the abnormalities of the circadian sleep/wake and of the pituitary function (Pellecchia et al. The GH-IGF system in amyotrophic lateral sclerosis: Correlations between pituitary GH secretion capacity, insulin-like growth factors and clinical features. Eur J Neurol. 2010;17(5):666-71. doi: 10.1111/j.1468-1331.2009.02896.x) which were revealed in ALS and FTLD. Please add some comments on this point.

The link between neuroinflammation and TDP-43 pathology could be addressed. Among emerging findings in this regard, general depletion of Cyclophilin A exacerbates protein aggregation, although inhibition of Cyclophilin A extra-cellular function, by reducing neuroinflammation, may affect TDP-43 pathology reducing level of insoluble phosphorylated TDP-43 in the spinal cord of the treated mice (Pasetto et al., J Neurosci. 2017;37:1413-1427). With regard to astrocytes role in ALS pathogenesis, cellular models revealed that astrocytes may have a neuroprotective role in seeded aggregation within motor neurons by reducing (mislocalized) cytoplasmic TDP-43, TDP-43 aggregation and cell toxicity (Smethurst et al., Brain 2020;143:430-440). If you agree, please address this issue and these mechanisms in the review context.

Author Response

Reviewer #1

We deeply appreciate the reviewer’s comments. Our response to each comment has been shown below.

p.5, lines 178-180 “Induction of patient-derived TDP-43 seeds resulted in spreading of TDP-43 aggregates in a cell-to-cell and in a corticofugal manner for SH-SY5Y cells and mice with a mutated human TDP-43 NLS (CamKIIa-hTDP-43NLSm), respectively”. This corticofugal spreading model (Braak et al., Nat Rev Neurol. 2013, 9, 708-714; Brettschneider J. et al., Ann Neurol. 2013, 74, 20-38) has been also confirmed in vivo by neuroimaging studies using structural (Schmidt et al., Neuroimage 2017;124:762-769; Müller et al., J Neurol Neurosurg Psychiatry 2016;87:570-9; Trojsi et al., PLoS One 2015;10:e0119045) and functional (Schulthess et al., Sci Rep. 2016;6:38391) neuroimaging techniques. A brief reference to this point could add emerging clinical information to the topic.

Answer: We also consider this point to be important in clinical aspects. The revised manuscript has addressed neuroradiological and neurophysiological evidences regarding to this context (Section5 in the page 5).

p.8, lines 249-252 With regard to the impaired clearance of TDP-43, the Authors reported that "haploinsufficiency of C9orf72 correlated to disruption of autophagy, particularly of endosomal trafficking, and the hippocampi and frontal cortices of C9orf72-mutated patients frequently and broadly displayed granulovacuolar degeneration”. However, the hypothalamic-pituitary axis may show TDP-43 aggregates and dipeptide repeat protein (DPR) inclusions in C9orf72-related ALS and/or FTLD-TDP cases (Dedeene L, et al. Circadian sleep/wake-associated cells show dipeptide repeat protein aggregates in C9orf72-related ALS and FTLD cases. Acta Neuropathol Commun. 2019;7(1):189. doi: 10.1186/s40478-019-0845-9). This evidence is related to the abnormalities of the circadian sleep/wake and of the pituitary function (Pellecchia et al. The GH-IGF system in amyotrophic lateral sclerosis: Correlations between pituitary GH secretion capacity, insulin-like growth factors and clinical features. Eur J Neurol. 2010;17(5):666-71. doi: 10.1111/j.1468-1331.2009.02896.x) which were revealed in ALS and FTLD. Please add some comments on this point.

Answer: It is of importance when considering extension of TDP-43 pathology in the non-motor systems. We have added the suggested point into the revised manuscript (Section 5 in the page 5).

The link between neuroinflammation and TDP-43 pathology could be addressed. Among emerging findings in this regard, general depletion of Cyclophilin A exacerbates protein aggregation, although inhibition of Cyclophilin A extra-cellular function, by reducing neuroinflammation, may affect TDP-43 pathology reducing level of insoluble phosphorylated TDP-43 in the spinal cord of the treated mice (Pasetto et al., J Neurosci. 2017;37:1413-1427). With regard to astrocytes role in ALS pathogenesis, cellular models revealed that astrocytes may have a neuroprotective role in seeded aggregation within motor neurons by reducing (mislocalized) cytoplasmic TDP-43, TDP-43 aggregation and cell toxicity (Smethurst et al., Brain 2020;143:430-440). If you agree, please address this issue and these mechanisms in the review context.

Answer: We agree the importance of neuroinflammatory in ALS pathogenesis. We have added some writings into the revised version (Section 7 in the page 8).

Reviewer 2 Report

I a m neurologist and neurophysiologist, but not a specialist in protein and their location in neurons. I know, that the first change in ALS is cortical inhibition (SICI) and that first change in peripheral motoneuron are fasciculations. But both facts were not mentioned. If this article will be read by clinicians, then both facts would be important to mention.

And- too many abbreviations. Then, some list of abbreviation would be helpful.

Author Response

We deeply appreciate the reviewer’s comments. Our response to each comment has been shown below.

I am neurologist and neurophysiologist, but not a specialist in protein and their location in neurons. I know, that the first change in ALS is cortical inhibition (SICI) and that first change in peripheral motoneuron are fasciculations. But both facts were not mentioned. If this article will be read by clinicians, then both facts would be important to mention.

Answer: We agree these points of importance. We have added TMS findings of ALS patients into the revised manuscript (Section 5 in the page 5).

And- too many abbreviations. Then, some list of abbreviation would be helpful.

Answer: We have added a list of abbreviation in the end of main text.